# Chromatin Remodelling Molecule ARID1A Determines Metastatic Heterogeneity in Triple-Negative Breast Cancer by Competitively Binding to YAP

**DOI:** 10.3390/cancers15092447

**Published:** 2023-04-25

**Authors:** Ye Wang, Xinyu Chen, Xiaosu Qiao, Yizhao Xie, Duancheng Guo, Bin Li, Jianing Cao, Zhonghua Tao, Xichun Hu

**Affiliations:** Department of Breast and Urologic Medical Oncology, Fudan University Shanghai Cancer Center, 270, Dong’an Road, Xuhui District, Shanghai 200032, China

**Keywords:** triple-negative breast cancer, metastasis, ARID1A, YAP complex

## Abstract

**Simple Summary:**

Heterogeneity is a prominent characteristic of triple-negative breast cancer, and it remains the predominant cause of treatment failure. ARID1A, a key subunit of the nuclear SWI/SNF protein complex, plays a crucial role in determining tumour identity by manipulating gene expression. The present study aims to investigate whether and how ARID1A contributes to the metastatic heterogeneity of triple-negative breast cancer.

**Abstract:**

Heterogeneity represents a pivotal factor in the therapeutic failure of triple-negative breast cancer (TNBC). In this study, we retrospectively collected and analysed clinical and pathological data from 258 patients diagnosed with TNBC at the Fudan University Cancer Hospital. Our findings show that low ARID1A expression is an independent prognostic indicator for poor overall survival (OS) and recurrence-free survival (RFS) in TNBC patients. Mechanistically, both nuclear and cytoplasmic protein analyses and immunofluorescent localisation assays confirm that ARID1A recruits the Hippo pathway effector YAP into the nucleus in human triple-negative breast cancer cells. Subsequently, we designed a YAP truncator plasmid and confirmed through co-immunoprecipitation that ARID1A can competitively bind to the WW domain of YAP, forming an ARID1A/YAP complex. Moreover, the downregulation of ARID1A promoted migration and invasion in both human triple-negative breast cancer cells and xenograft models through the Hippo/YAP signalling axis. Collectively, these findings demonstrate that ARID1A orchestrates the molecular network of YAP/EMT pathways to affect the heterogeneity in TNBC.

## 1. Introduction

Triple-negative breast cancer (TNBC) accounts for approximately 15–20% of all breast cancers and is widely considered to have the worst prognosis due to its high degree of malignancy and limited treatment options [1,2]. Several targets have been explored for TNBC treatment, including phosphatidylinositol 3 kinase/protein kinase B/mammalian target of rapamycin (PI3K/AKT/mTOR), trophoblast membrane antigen 2 (TROP2), programmed cell death receptor 1 (PD1), and programmed cell death receptor ligand 1 (PD-L1); however, their efficacy is limited [3,4]. The heterogeneous nature of the disease is believed to contribute to the limited effectiveness of targeted agents, highlighting the importance of identifying valuable markers.

The mammalian chromatin remodelling complex SWI/SNF is a large molecular complex composed of multiple subunits, of which the core member, ARID1A, is primarily involved in the transcriptional regulation of transcription factors, cellular signalling, and epigenetic modifications [5,6]. Previous studies have shown that ARID1A expression is associated with ovarian cancer [7]. A recent study also demonstrated that ARID1A mutations predict the efficacy of immunotherapy [8]. Accumulated evidence over the past decade has also shown that ARID1A is a predictive factor for treatment response to tamoxifen, fulvestrant, and 5-FU, and influences HDAC1/BRD4 activity [9]. In luminal A breast cancer, ARID1A downregulation induces EMT, suggesting that it may be a target in ER-positive breast cancer. However, the role of ARID1A in the metastasis and progression of TNBC remains unclear.

Recent studies have revealed that breast cancer is frequently accompanied by mutations in PIK3CA, Akt, and TP53, but still around 10% of patients with breast cancer have ARID1A mutations [10]. Our previous research has indicated that the absence of ARID1A is associated with bone metastasis in TNBC [11], underscoring the role of ARID1A in the invasive metastasis of TNBC. To deepen our understanding of the role of ARID1A in TNBC, we collected specimens from breast cancer patients and investigated its impact on the prognosis of TNBC patients. Building on this foundation, we explored the linkage between ARID1A and various biological processes, identified interacting proteins, and identified potential sites of action that ARID1A might play in cellular functions. Our data suggest that ARID1A competes with YAP by binding to it, thereby inhibiting the formation of TEAD/YAP complexes, and affecting tumour metastasis. These findings are crucial in understanding the mechanism of ARID1A in TNBC and developing precise treatments.

## 2. Materials and Methods

### 2.1. Patient Samples

Samples from 258 TNBC patients who underwent surgery with no prior systemic neoadjuvant treatment between August 2015 and December 2017 at the Fudan University Shanghai Cancer Center (FUSCC) (Shanghai, China) were collected. All the patients were histologically diagnosed by an expert pathologist. Overall survival (OS) was defined as the time between the date of surgery and the date of death or last follow-up. Disease-free survival (DFS) was defined as the time between the date of surgery and the date of a reported event such as death, locoregional recurrence, contralateral breast cancer, distant metastasis, second malignancy, or the date of the last follow-up. Written informed consent was obtained from all patients and all research protocols were approved by the FUSCC Medical Ethics Committee.

### 2.2. Cell Lines and Cell Culture

To investigate the role of ARID1A in TNBC, we used human breast cancer cell lines MDA-MB-231 and BT-549, and murine 4T1 in our experiments. Human breast cancer cell lines MDA-MB-231, BT-549 and murine 4T1 and HEK293T cells were purchased from ATCC. BT-549 and 4T1 cells were cultured in RPMI-1640 medium, whereas MDA-MB-231 and HEK293T cells were cultured in DMEM (Thermo Fisher Scientific, Waltham, MA, USA). All of the medium was supplemented with 10% foetal bovine serum (FBS, Gibco, Waltham, MA, USA) and 1% pen–strep antibiotic (Beyotime, Shanghai, China), and all cells were cultured at 37 °C in a humidified atmosphere containing 5% CO_2_. The cell morphology was observed under an optical microscope.

### 2.3. Lentiviral shRNA Production, Infection and Knockdown of Genes

To generate ARID1A knockouts in MDA-MB-231 and BT549 cells, the Lenti-CAS9-Puro and GV371-EGFP vectors (Shanghai Genechem Co., Ltd., Shanghai, China) were sequentially transfected. Similarly, to knock out ARID1A in 4T1 cells, the Lenti-CAS9-Puro vector and GV493-EGFP vector (Shanghai Genechem Co., Ltd., Shanghai, China) were sequentially transfected. For overexpression of ARID1A, HEK293T cells were transfected with the appropriate plasmids using Lipofectamine 3000 transfection reagent (L3000-150; Invitrogen, Waltham, MA, USA), and the virus-containing supernatant was collected after 72 h of incubation. The virus was then concentrated and transfected into TNBC cells using Polybrene (sc-134220; Santa Cruz Biotechnology, Santa Cruz, CA, USA). Transfected cells were selected 7 days later with high concentrations of neomycin until the cells were no longer dying, and low neomycin concentrations were subsequently maintained. To knock down YAP expression, shRNA (Shanghai Genechem Co., Ltd., Shanghai, China) was transfected into MDA-MB-231 cells. Similarly, MDA-MB-231 cells were transfected with BRG1 siRNA (J-010431-06-0005; Horizon, Cambridge, UK) to knock down the expression of BRG1 using Lipofectamine 3000 reagent (L3000-150; Invitrogen).

The shRNA sequences were as follows:


**Species**

**Gene**

**Sequences**
mouseARID1A^KO^-15′-GGTCCCTGTTGTTGCGAGTA-3′mouseARID1A^KO^-25′-gACCCCATGACCATGCAGGGC-3′mouseYAP^KD^-15′-AGGCCAGTACTGATGCAGGTA-3′mouseYAP^KD^-25′-CAGGACCTCTTCCTGATGGAT-3′humanARID1A^KO^-15′-CACCGATGGTCATCGGGTACCGCTG-3′humanARID1A^KO^-25′-CACCGCCCCTCAATGACCTCCAGTA-3′humanARID1A^OE^-15′-CACCGGGCGCTCTAGCCGCTCAGTC-3′humanARID1A^OE^-25′-CACCGCTTGGGTCGAGGCTGCTGCG-3′humanYAP^KD^-15′-CCCAGTTAAATGTTCACCAAT-3′humanYAP^KD^-25′-CACCAAGCTAGATAAAGAA-3′

### 2.4. Cellular Fractionation, Western Blot and Co-IP Assay

Cellular fractions were isolated using the Nuclear and Cytoplasmic Protein Extraction Kit (20126ES50; Yeasen, Shanghai, China). The cells were washed with 1× PBS and lysed in RIPA lysis buffer containing protease inhibitors, and the protein levels in the lysates were quantified using an enhanced bicinchoninic acid assay (BCA) kit (P0012, Beyotime, Shanghai, China). Cell lysates were separated by sodium dodecyl sulfate-polyacrylamide gel electrophoresis, transferred to PVDF membranes, and immunoblotted with the indicated antibodies. For immunoprecipitation assays, protein lysates were incubated with anti-ARID1A, anti-YAP, anti-BRG, anti-TEAD, anti-Flag, or normal IgG antibody at 4 °C overnight with rotation. On day 2, the mixture was incubated with protein A/G beads at room temperature for 2–3 h. After washing three times with lysis buffer, the beads were boiled and subjected to Western blotting. The antibodies used were as follows: ARID1A (ab272905, Abcam, Cambridge, UK); YAP (#14074, CST, Fall River, MA, USA); p-YAP (#13008, CST); BRG1 (ab110641, Abcam); TEAD (ab133533, Abcam); E-cadherin (#14472, CST); N-cadherin (#13116, CST); MMP2 (ab92536, Abcam); MMP-9 (ab283575, Abcam); Snail (#3879, CST); Vimentin (ab92547, Abcam); GAPDH (60004-1-Ig, Proteintech, Chicago, IL, USA); Histone H3 (ab17917, Abcam), Flag (20543-1-AP, Proteintech).

### 2.5. Preparation of RNA and Quantitative Real-Time PCR

To analyse gene expression levels, total RNA was extracted from the cells using the TRIzol reagent (Invitrogen) following the manufacturer’s instructions. RNA was then reverse transcribed and subjected to real-time PCR using SYBR Premix Ex Taq (TaKaRa Bio, Tokyo, Japan) with triplicate runs. The specificity of the reaction was confirmed by melting curve analysis at the dissociation stage. The relative quantitative method was used for analysis, with the average ΔCt of untreated cells serving as the calibrator. The endogenous control was GAPDH. The primer sequences used in the PCR are listed in Appendix A.

### 2.6. RNA-Seq Library Construction and Analysis

The MDA-MB-231 cells were used to extract total RNA at a quantity of 1 μg per sample. RNA-seq libraries were prepared using the VAHTS mRNA-seq V2 library prep kit for Illumina (Vazyme, Nanjing, China), as per the manufacturer’s instructions. The libraries were sequenced on the Illumina sequencing platform using a 150 bp paired-end run, following quality inspection with FastQC (http://www.bioinformatics.babraham.ac.uk/projects/fastqc (accessed on 21 November 2021)). The spliced read aligner HISAT2, supplied by the Ensembl Human Genome Assembly (Genome Reference Consortium GRCh38), was used to align the clean reads against the reference genome. The gene expression levels were calculated as Reads Per Kilobase Million (RPKM). The RNA-seq data is currently being uploaded to the Gene Expression Omnibus (GEO) Repository.

### 2.7. Wound Healing Assay

The cells were seeded in 6-well plates at a final density of 100% and cultured for 24 h. After this, the cell surface was scratched with a pipette tip and washed three times with PBS. The cells were then cultured in a humid incubator with 5% CO_2_ at 37 °C. At 0, 12, and 24 h, the cells were removed, observed under a microscope, and photographed. The mean intercellular distances were calculated using ImageJ software.

### 2.8. Transwell Migration and Invasion Assays

To assess the migration ability of the cells, a 24-well transwell plate (Corning Inc., Corning, NY, USA) was used, while the tumour cell invasion assay utilised Matrigel (BD Bioscience, Franklin Lakes, New Jersey, USA) added to the transwell chambers. Cells were seeded into the upper chamber of the 24-well plate, resuspended in RPMI with serum-free medium, and the lower chamber was filled with medium containing 10% FBS. After incubation for 48 h, the cells were fixed in 4% paraformaldehyde for 15 min, stained with crystal violet, and counted using an inverted light microscope (Olympus, Tokyo, Japan).

### 2.9. Immunofluorescent Staining

The cells were cultured in 20 mm confocal dishes until they reached 50% confluence after 24 h. They were then washed with PBS and fixed in 4% paraformaldehyde for 30 min. Subsequently, the cells were permeabilised with 0.5% Triton X-100 solution for 20 min and then blocked with 5% bovine serum albumin for 1 h. Primary antibodies (YAP 1:200, No.13584-1-AP, Proteintech, Wuhan, China) were added and incubated overnight at 4 °C. Finally, the images were acquired using a confocal laser scanning microscope (Leica, Wetzlar, Germany).

### 2.10. In Vivo Tumour Metastasis Model

Six- to seven-week-old female BALB/c mice were procured from Shanghai Model Organisms Center, Inc. To conduct the in vivo assay, 1 × 10^5^ 4T1 cells (NC, ARID1A^KO^, YAP^KO^, and ARID1A/YAP^DKO^) were inoculated into the tail veins of female mice. After five weeks, the mice were sacrificed, and their lungs were fixed in 4% paraformaldehyde solution for IHC staining. All animal experiments were conducted in accordance with the ethical guidelines of the Animal Care Committee of FUSCC.

### 2.11. Immunohistochemical Analysis

The tumours were fixed with 4–10% paraformaldehyde for 24 h at 4 °C and then dehydrated through a series of ethanol concentrations (70–100%). Subsequently, the tumours were immersed in xylene and embedded in paraffin. Sections 5 µm thick were obtained from the embedded samples. The sections were then de-paraffinised, and endogenous peroxidase was blocked with 0.3% H_2_O_2_ in distilled water for 20 min. To retrieve the antigen, the sections were boiled in citrate buffer for 1 min. Non-specific reactions were blocked by incubating the sections with foetal bovine serum in PBS for 1 h. After protein blocking, the sections were incubated overnight at 4 °C with ARID1A (#12354, CST), E-cadherin (#14472, CST), N-cadherin (#13116, CST), and Snail (ab180714, Abcam) antibodies in PBS. Corresponding secondary antibodies were added to the sections for 1 h at room temperature. After rinsing with PBS, the slides were stained with 3,3′-diaminobenzidine and counterstained with haematoxylin. Images were captured using an inverted light microscope (Olympus).

### 2.12. Hematoxylin/Eosin Staining

The lung tissues from the mice were fixed in 4–10% paraformaldehyde for 24 h at 4 °C. Subsequently, the tissues were dehydrated using 70–100% ethanol, immersed in xylene, and embedded in paraffin. Once embedded, 5-µm-thick sections were obtained and deparaffinised using xylene. The sections were then hydrated with decreasing concentrations of alcohol, stained with haematoxylin and eosin solution, and dehydrated in pure alcohol. Finally, the slides were made transparent using xylene and viewed under a microscope (Olympus).

### 2.13. Statistical Analysis

Statistical analysis was conducted using GraphPad Prism 9.0 (GraphPad Software, Boston, MA, USA). The data are presented as means with standard error of the mean (SEM). Unpaired Student’s *t*-test, one-way analysis of variance (ANOVA), or two-way ANOVA were performed for quantitative data as appropriate. To evaluate the effect of prognostic factors on patient survival, Kaplan–Meier curves were used. A *p*-value < 0.05 was considered statistically significant in all cases.

## 3. Results

### 3.1. ARID1A-Low Expression Associated with Metastasis and Poor Prognosis in TNBC

According to data from the Cbioportal database, approximately 10% of all cancer cases and 7% of breast cancer cases exhibit ARID1A inactivation, including truncating mutations and deep depletion (as depicted in Figure 1A). To investigate the potential prometastatic role of ARID1A in triple-negative breast cancer (TNBC), we first analysed the Kaplan–Meier plotter and BC gene expression miner and observed that ARID1A expression was significantly associated with overall survival (OS), relapse-free survival (RFS), and distant metastasis-free survival (DMFS), as illustrated in Figure 1B,C. To further investigate the effects of ARID1A on TNBC prognosis, we evaluated ARID1A protein levels by conducting immunohistochemical analysis on a TNBC cohort comprising 258 patients diagnosed at the Fudan University Shanghai Cancer Center (FUSCC) between August 2015 and May 2016 (as presented in Figure 1D and Appendix A). Our analysis revealed that TNBC patients with higher ARID1A expression levels demonstrated significantly better OS and RFS, as shown in Figure 1E.

In addition, we analysed the mRNA and protein expression of ARID1A in different subtypes of breast cancer and found that ARID1A levels in the luminal and HER2-positive subtypes were higher than those observed in TNBC (as demonstrated in Figure 1F). This finding suggests that ARID1A is negatively associated with metastatic potential. Taken together, our findings indicate that ARID1A inhibition promotes TNBC metastasis.

### 3.2. Inhibition of ARID1A Promotes Migration and Invasion in TNBC Cells

To determine the role of ARID1A in regulating TNBC metastasis, MDA-MB-231 and BT-549 cells were silenced by CRISPR/Cas9-mediated (ARID1A^KO^) and overexpressed by the plasmid (ARID1A^OE^). The efficiency of the knockout and overexpression was evaluated by qRT-PCR and Western blotting (Figure 2A and Appendix A). We first captured the morphological changes in MDA-MB-231 and BT-549 cells after ARID1A expression was manipulated. Compared with the wild-type, the loss of ARID1A in MDA-MB-231 and BT-549 cell lines resulted in significant pseudopodia (Figure 2B and Appendix A), which contributed to the migration of cancer cells. Consistently, ARID1A silencing promoted cell migration and invasion in transwell and wound healing assays, whereas ARID1A overexpression decreased cell migration and invasion (Figure 2C,D and Appendix A). Cell proliferation contributes to tumour growth and might affect the results of tumourigenicity in experiments. To exclude the effect of cell proliferation on clonal growth, we re-analysed the data and attempted to standardise clonal growth with the proliferation rate. The results showed that the increased number of metastasis clones induced by ARID1A^KO^ was not affected by the proliferation (Appendix A).

### 3.3. ARID1A Inhibits TNBC Metastasis In Vivo

As an alternative experiment to the orthotopic transplantation tumour model, the vein model was used in this study to assess the regulatory effect of ARID1A on TNBC cell metastasis ability by simulating the second half of the metastasis process. We first knocked out ARID1A in 4T1 cells using CRISPR/Cas9 (Appendix A). ARID1A^KO^-1 was used in subsequent animal experiments because it had the best knockout efficiency. Next, we established a mouse xenograft model of lung metastasis by injecting the mice with 4T1 cells (Figure 3A). On day 35 after injection, six out of seven mice with ARID1A^KO^ tumours developed lung metastasis, whereas only two out of seven mice developed lung metastasis in the ARID1A^NC^ group. The difference in the metastatic ratio between the two groups was significant (Figure 3B). Moreover, the number and area of tumour foci in the lungs, marked with fluorescence, were also notably increased in the ARID1A^KO^ group compared to those in the ARID1A^NC^ group (Figure 3C,D). We also performed haematoxylin and eosin staining of the lung tissue to further verify the differences in lung metastasis. Not surprisingly, the lung tissue in the ARID1A^KO^ group lost its alveolar structure and was morphologically disturbed, with significant tumour formation compared to that in the ARID1A^NC^ group (Figure 3E). Moreover, we analysed 258 patients with TNBC with metastases and found that those with low ARID1A expression were more likely to have lung, liver, and bone metastases (Appendix A). Collectively, we believe that ARID1A can suppress tumour metastasis in TNBC.

### 3.4. ARID1A Regulates EMT in TNBC

The morphological changes after ectopic differentiated expression of ARID1A in TNBC cells (Figure 2B and Appendix A) suggest that ARID1A suppresses TNBC metastasis by regulating the epithelial–mesenchymal transition (EMT). To verify this, we first detected EMT markers in the cells by Western blotting. The results show that ARID1A silencing decreased E-cadherin expression and enhanced N-cadherin, MMP-2, MMP-9, and Snail expression in MDA-MB-231 and BT-549 cells. Contrary results were found in ARID1A over-expressed MDA-MB-231 and BT-549 cells (Figure 4A,B). Next, we performed an IHC assay to evaluate the markers of EMT in TNBC patients and found that the protein levels of E-cadherin in ARID1A-low patients were significantly lower than that in ARID1A-high patients, whereas N-cadherin and Snail were dramatically higher in ARID1A-low patients (Figure 4C,D). These results revealed that ARID1A regulates EMT and that ARID1A-mediated TNBC metastasis may be associated with EMT regulation.

### 3.5. YAP and Hippo Pathway Involved in ARID1A-Regulated EMT and Tumour Metastasis

To understand the effect of ARID1A on TNBC metastasis, RNA sequencing analysis was performed to compare gene expression profiles in MDA-MB-231 cells treated with ARID1A^KO^ or ARID1A^OE^ and to analyse differential gene expression (DGEs). Eleven genes were identified as common differential genes, and the ECM-receptor interaction and Hippo signalling pathways were enriched (Figure 5A,B). Because it is closely associated with EMT regulation, we focused on how the Hippo pathway is involved ARID1A-associated EMT in our study. First, we analysed the DGEs related to the Hippo pathway and found that some genes downstream of YAP showed significant changes in TNBC cells between wild-type and ARID1A^OE^ groups (Figure 5C). However, the protein levels of YAP were not significantly different between the groups (Figure 5D and Appendix A). These results suggest that YAP, a crucial regulator of the Hippo pathway, is involved in ARID1A regulation of EMT. Next, we generated YAP knockdown (YAP^KD^), ARID1A knockout, and double knockdown cells (ARID1A/YAP^DKO^) to verify the role of YAP in ARID1A-associated EMT and tumour metastasis (Figure 5E). The results showed that the knockdown of YAP expression in MDA-MB-231 cells significantly decreased migration and invasion, while YAP silencing in ARID1A knockout MDA-MB-231 cells markedly reversed the increased migration and invasion (Figure 5F,G). Finally, we found that YAP knockdown resulted in an increase in E-Cad in MDA-MB-231 cells, whereas N-Cad, Vimentin and Snail decreased. Interestingly, YAP silencing in ARID1A knockout MDA-MB-231 cells, the upregulated N-Cad, Vimentin, and Snail were notably reversed, vice versa, ARID1A^KO^ induced expression inhibition of E-Cad was relieved with significantly high expression (Figure 5H). Taken together, these results suggest that YAP and the Hippo pathway are involved in ARID1A associated EMT and tumour metastasis.

### 3.6. ARID1A Competitively Binds YAP to Form ARID1A/YAP Complex

These results show that ARID1A activates YAP and the Hippo pathway, but not via the regulation of YAP expression. To explore the mechanism by which ARID1A regulates YAP and the Hippo pathway, we measured the YAP protein expression in the cytoplasm and nucleus. Surprisingly, the expression of YAP and p-YAP increased in the cytoplasm and decreased in the nucleus when ARID1A was silenced (Figure 6A and Appendix A), and vice versa (Figure 6B and Appendix A). Immunofluorescence assays showed a significant decrease in nuclear YAP expression after the knockdown of ARID1A (Figure 6C). These results indicate that ARID1A promotes the entry of YAP into the nucleus.

To further investigate the relationship between ARID1A and YAP in the nucleus, Co-IP showed that the combination of ARID1A and YAP decreased after silencing ARID1A, and overexpression of ARID1A could increase their combination (Figure 6D and Appendix A). Previous studies have shown that YAP can form a complex with BRG1 and affect tumour progression [12], both ARID1A and BRG1 are components of the SWI/SNF complex. Therefore, we wanted to know whether the formation of the YAP/BRG1 complex is related to ARID1A. We observed the YAP/ARID1A/BRG1 interaction by Co-IP after the transfection of siBRG1 in MDA-MB-231 cells (Appendix A). We found that knockout of ARID1A reduced YAP/BRG1 binding, whereas reduced BRG1 expression had no significant effect on ARID1A/YAP binding (Figure 6E), suggesting that ARID1A plays an important role in the formation of the YAP/BRG1 complex. YAP, which is present in the nucleus, can bind to TEAD to form a complex that activates the expression of its downstream genes, thereby promoting tumour development [13,14]. When the expression of ARID1A is inhibited, YAP dissociated from the compound and bound to TEAD (Figure 6F). These results reveal that ARID1A competitively binds YAP to TEAD and affects the expression of the downstream pathway of the YAP-TEAD complex.

### 3.7. ARID1A Combined with YAP via the WW Domain

These studies have demonstrated that ARID1A can competitively bind YAP with TEAD. To investigate the specific site of action of ARID1A with YAP, we designed truncator plasmids with a FLAG tag and transfected these plasmids into 293T cells (Figure 7A). The results of co-immunoprecipitation (Co-IP) experiments showed that ARID1A can interact with YAP in the presence of the WW domain (Figure 7B,C and Appendix A). This suggests that the WW domain is the binding region of the ARID1A/YAP complex.

To further understand the effect of the ARID1A/YAP complex on tumour invasion, we transfected the truncated plasmid into MDA-MB-231 cells and conducted wound healing and transwell assays. The results confirmed that loss of the WW domain significantly increased the invasive ability of tumour cells (Figure 7D,E). Additionally, the deletion of the WW domain regulated the transcriptional ability of downstream genes E-cadherin, N-cadherin, and Snail (Figure 7F). Taken together, these findings suggest that ARID1A binds to the WW domain of YAP to form an ARID1A/YAP complex. This complex regulates the transcription of genes downstream of YAP, thereby inhibiting tumour invasion and metastasis.

### 3.8. ARID1A and YAP Co-Regulate Tumour Metastasis In Vivo

Our study confirmed that ARID1A inhibits the formation of the YAP/TEAD complex, which subsequently inhibits tumour metastasis in vitro. Therefore, we explored whether ARID1A and YAP influence tumour metastasis in vivo. In this experiment, mice were divided into control, ARID1A^KO^, YAP^KD^, and ARID1A/YAP^DKO^ groups by tail vein injection of different 4T1 cells (Figure 8A). As shown in Figure 8B, inhibition of ARID1A significantly promoted the formation of lung metastases compared to that in the other three groups, and simultaneous inhibition of ARID1A and YAP partially counteracted ARID1A-induced lung metastasis (Figure 8C). These findings were further confirmed by haematoxylin-eosin staining of the lung tissue, which was morphologically disturbed with significant tumour formation in the ARID1A^KO^ group and compared with the ARID1A^KO^ group; the alveolar structure and tumour formation were decreased in the ARID1A/YAP^DKO^ groups (Figure 8D). In conclusion, ARID1A and YAP co-regulate tumour metastasis in vivo.

## 4. Discussion

Breast cancer is the most prevalent type of malignancy worldwide. TNBC, the most malignant type of breast cancer, has a poor prognosis owing to its heterogeneity and lack of suitable targets. Although molecular classifications of TNBC have been established, including the luminal androgen receptor (LAR), immunomodulatory (IM), basal-like immune-suppressed (BLIS), and mesenchymal-like (MES) [15], the corresponding targeted therapies for each subtype are immature. Thus, it is important to identify the heterogeneity and novel therapeutic targets for TNBC treatment.

Previously, ARID1A was considered a critical suppressor of cancer progression in gastric, pancreatic, and other cancers [16,17], suggesting that ARID1A-inhibited tumour development is universal. As a core member of the SWI/SNF family, ARID1A is primarily associated with transcriptional regulation of transcription factors, cellular signalling, and epigenetic modifications, which have high-frequency mutations in human malignancies [18]. Several studies have found that ARID1A downregulation is associated with poor prognosis in luminal breast cancer and mediates treatment responses to tamoxifen and fulvestrant. In this study, we found that a subgroup of patients with TNBC with low ARID1A expression had a poorer prognosis. Subsequent in vitro and in vivo experiments confirmed that ARID1A is a critical factor in the progression of TNBC. First, we verified that ARID1A is a critical regulator of EMT in TNBC by competitively binding to YAP and regulating the Hippo pathway.

EMT is a key initial step in the invasive metastasis of tumours. When the EMT pathway is activated, epithelial cells lose polarity, contact with the surrounding and stromal cells is reduced, and intercellular interactions are reduced, allowing for increased cell migration and motility [19]. Recently, another study found that the downregulation of ARID1A could accelerate EMT in luminal breast cancer. In the present study, we also found that the downregulation of ARID1A expression resulted in EMT in TNBC cells, suggesting that ARID1A plays a role in regulating EMT in breast cancer and could be a useful biomarker for identifying a subgroup of TNBC patients with poor prognosis and a potential target for preventing the metastasis of TNBC.

Several molecules and signalling pathways are considered key regulators of EMT [20]. In the present study, the Hippo signalling pathway was found to be the top-ranked ARID1A-affected pathway, suggesting that ARID1A regulates EMT and tumour metastasis in TNBC by controlling the Hippo pathway. Previous studies have shown that the Hippo pathway is closely associated with EMT [21]. Recent studies have also indicated that YAP is a crucial regulator and effector of the Hippo signalling pathway, and is involved in proliferation, metastasis, and EMT regulation [22]. In our study, we found that the downstream molecules of YAP were affected by ARID1A; however, the mRNA and protein expression levels of YAP did not change after ARID1A downregulation and upregulation. YAP is normally located in the cytoplasm in an inactive state. Upon stimulation by the external environment, YAP translocates from the cytoplasm to the nucleus and is activated [23]. Li et al. found that cell phenotypes were closely related to differences in YAP nucleocytoplasmic co-localisation [24]. Surprisingly, we found that ARID1A could induce YAP to enter the nucleus in TNBC cells, suggesting that ARID1A could recruit and hijack YAP to regulate the YAP/Hippo pathway. The current dogma of YAP regulation is that YAP binds to TEAD to form the YAP/TEAD complex and directs gene expression changes that control a range of biological events [25,26]. In this study, we found that ARID1A competitively binds to YAP to inhibit the production of the YAP/TEAD complex, leading to the suppression of its transcriptional activity and the consequent inhibition of tumour metastasis. Mutations in ARID1A eliminate this inhibitory effect and promote EMT and metastasis.

In conclusion, we verified that ARID1A is a biomarker for determining the heterogeneity of metastasis in TNBC and discovered a novel regulatory link between ARID1A and YAP in TNBC. Our data argument in favour of a paradigm is that ARID1A affects tumour metastasis by inhibiting the production of the YAP/TEAD complex and regulating the EMT signalling pathway (Figure 9). Based on this regulatory link, the modulation of ARID1A expression or function could be a promising strategy for treating YAP-driven cancers.

## 5. Conclusions

Our study provides new insights into the previously unrecognised role of ARID1A in TNBC metastasis and may provide new therapeutic options for patients with TNBC.

## Figures and Tables

**Figure 1 cancers-15-02447-f001:**
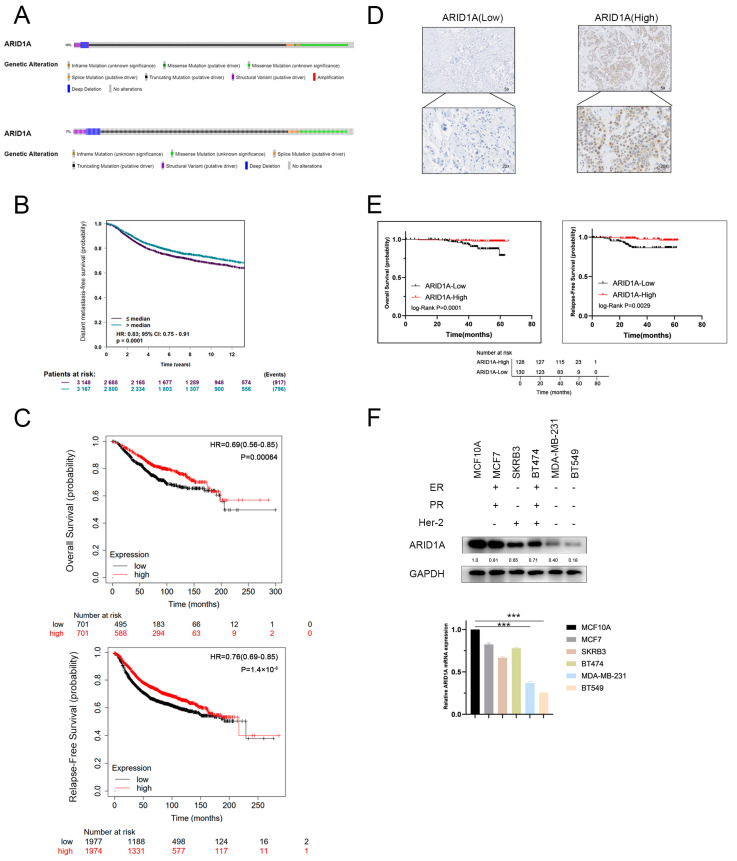
ARID1A-low expression associated with metastasis and poor prognosis in TNBC. (**A**) Cbioportal analysis of the expression of ARID1A in cancers. Approximately 10% of cases of cancers (above) and 7% of cases of breast cancer (below) have inactivation of ARID1A, including truncating mutation, deep depletion, and missense mutation. (**B**) BC gene expression miner analysis of DMFS in TNBC patients with low and high expression of ARID1A. (**C**) Kaplan–Meier analysis of OS and RFS in TNBC patients with low and high expression of ARID1A. (**D**) Immunohistochemistry for ARID1A (brown) in the breast cancer tissue. (**E**) The OS and RFS of 258 TNBC patients with low and high expression of ARID1A. (**F**) The mRNA and protein expression of ARID1A in different subtypes of breast cancer. Error bars represent the standard deviation among replicates. Data are presented as the mean ± SEM (n = 3), *** *p* < 0.05. The uncropped blots are shown in Appendix A.

**Figure 2 cancers-15-02447-f002:**
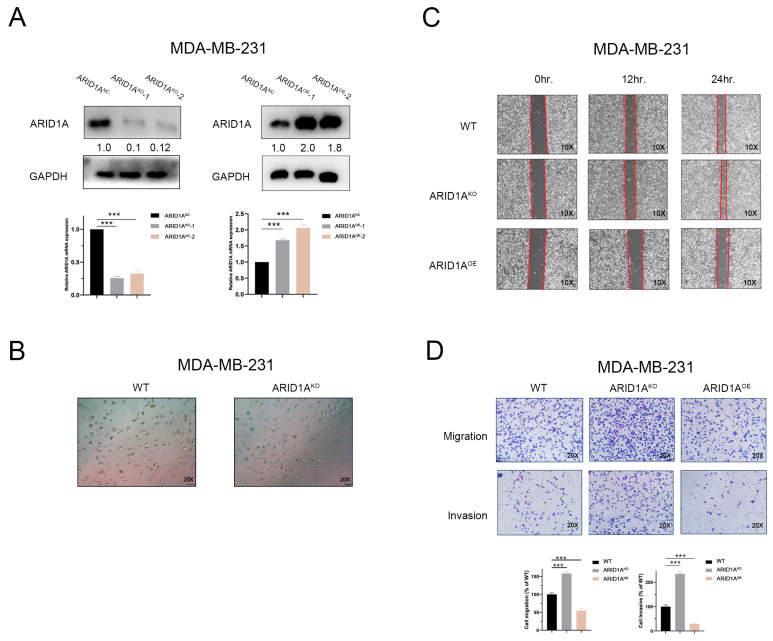
Inhibition of ARID1A promotes migration and invasion in TNBC cells. (**A**) MDA-MB-231 cells were silenced by CRISPR/Cas9-mediated and over-expressed by the plasmid. ARID1A expression levels were determined by Western blotting and qRT-PCR. (**B**) Cell morphology observed under an optic microscope in MDA-MB-231 of each group. (**C**,**D**) Migration and invasion of MDA-MB-231 cells of each group detected by wound healing assay (**C**), transwell migration assay and transwell invasion assay (**D**). Error bars represent the standard deviation among replicates. Data are presented as the mean ± SEM (n = 3), *** *p* < 0.05. The uncropped blots are shown in Appendix A.

**Figure 3 cancers-15-02447-f003:**
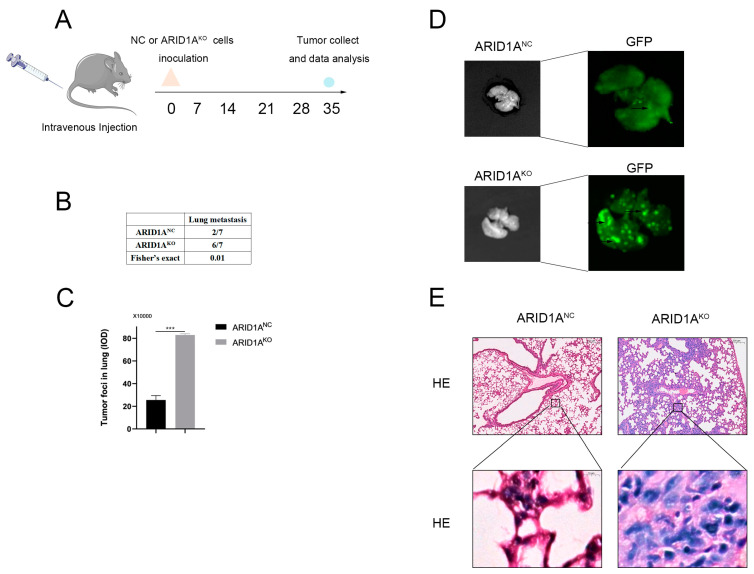
ARID1A inhibits TNBC metastasis in vivo. (**A**) Schematic diagram of a migration tumour model. (**B**) Incidence of lung metastasis in ARID1A^NC^ and ARID1A^KO^ cells. (**C**) Quantitation of tumour foci (IOD of fluorescent area) in the lung of mice injected with ARID1A^NC^ and ARID1A^KO^ cells. (**D**) Images showing the colonisation and outgrowth foci of ARID1A^NC^ and ARID1A^KO^ cells in the lung of recipient mice. (**E**) Hematoxylin and eosin (H&E) staining analysis of tissue sections from recipient mice at 5 weeks after transplantation. Scale bar, 200 μm (up) and 10 μm (down). Error bars represent the standard deviation among replicates. Data are presented as the mean ± SEM (n = 3), *** *p* < 0.05.

**Figure 4 cancers-15-02447-f004:**
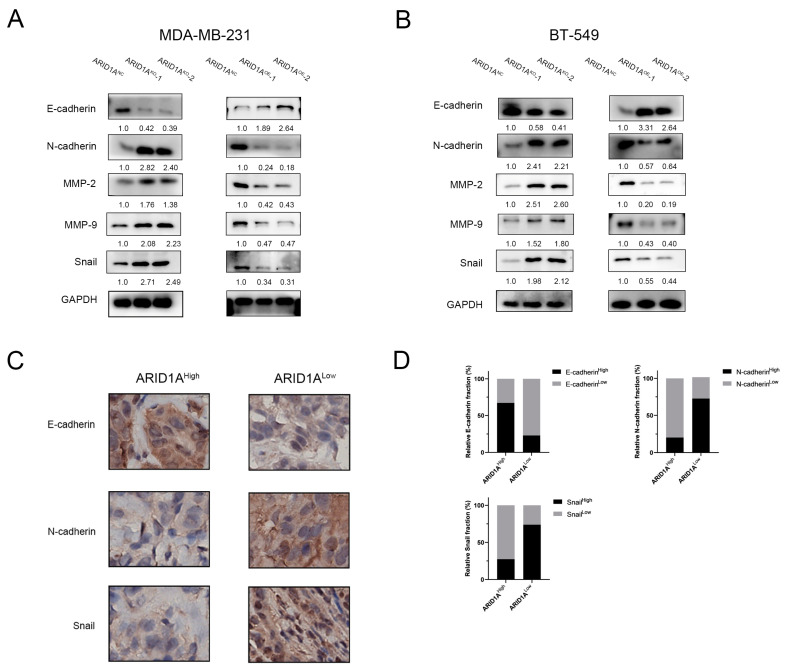
ARID1A regulates EMT in TNBC. (**A**) Western blot assay of EMT markers expression in each group of MDA-MB-231 cells. (**B**) Western blot assay of EMT markers expression in each group of BT-549 cells. (**C**) Immunohistochemistry for E-cadherin, N-cadherin and snail in the breast cancer tissue. Scale bar, 10 μm. (**D**) Relative fraction of E-cadherin, N-cadherin and Snail expression in 258 TNBC patients. Error bars represent the standard deviation among replicates. Data are presented as the mean ± SEM (n = 3). The uncropped blots are shown in Appendix A.

**Figure 5 cancers-15-02447-f005:**
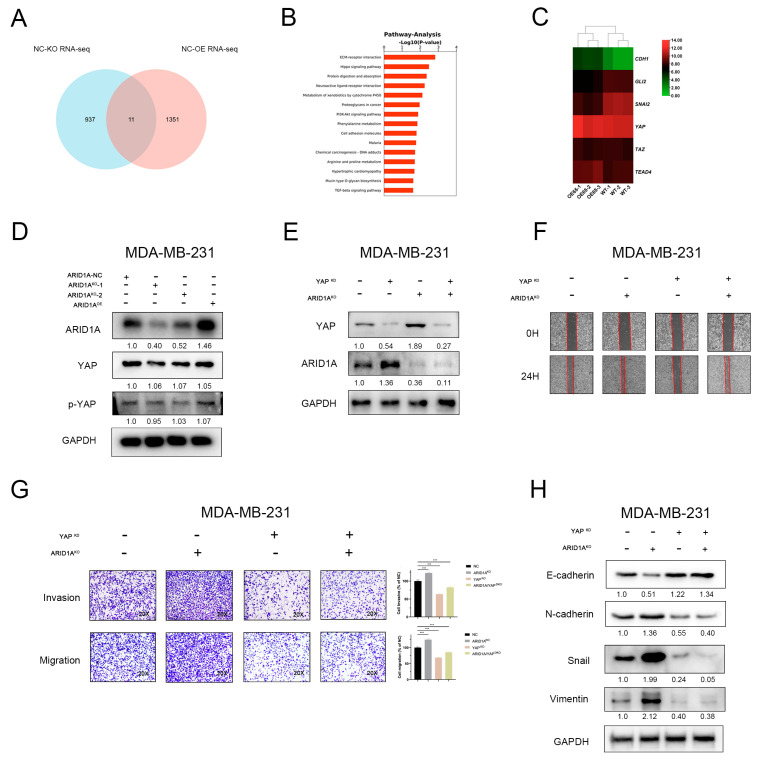
YAP and Hippo pathway involved in ARID1A-regulated EMT and tumour metastasis. (**A**) RNA sequencing analysis was performed to compare gene expression profiles in each group. (**B**) Pathway enrichment analysis of differential gene expression profiles in RNA-seq. The degree of Pathway enrichment is represented by the −Log 10 *p*-Value and the number of transcripts enriched in each category. (**C**) Heatmap of Hippo pathway changes in TNBC cells between wild type and ARID1A^OE^ groups. (**D**) Western blot assay of YAP expression in each group of MDA-MB-231 cells. (**E**) Separate or combined knockdown of ARID1A and YAP in MDA-MB-231 cells, the expression levels were determined by Western blotting. (**F**,**G**) Migration and invasion of MDA-MB-231 cells of each group detected by wound healing assay (**F**), transwell migration assay and invasion assay (**G**). (**H**) Western blot assay of EMT markers expression in each group of MDA-MB-231 cells. Error bars represent the standard deviation among replicates. Data are presented as the mean ± SEM (n = 3), *** *p* < 0.05. The uncropped blots are shown in Appendix A.

**Figure 6 cancers-15-02447-f006:**
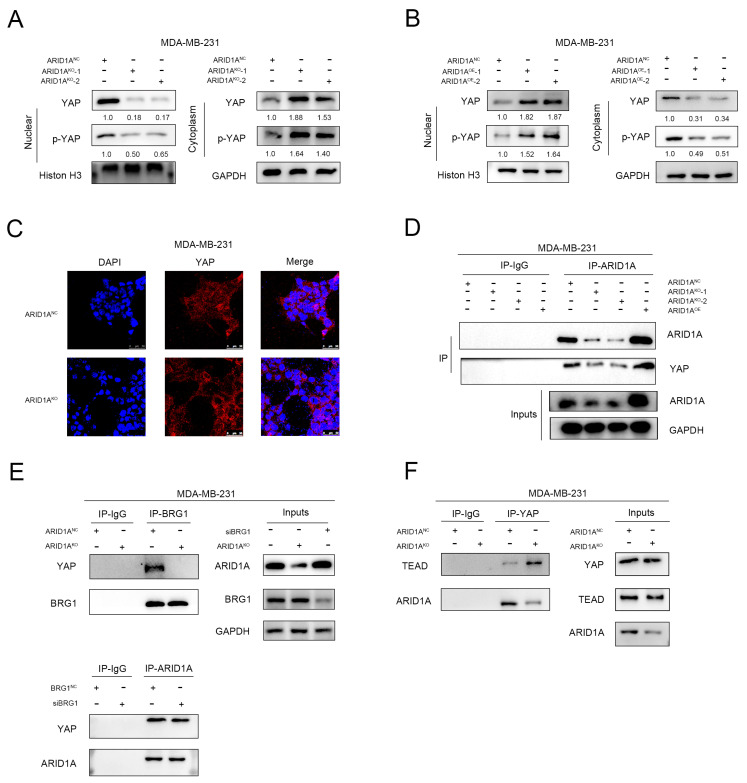
ARID1A competitively binds YAP to form ARID1A/YAP complex. (**A**) Western blot assay of cytoplasmic and nuclear YAP and p-YAP expression in MDA-MB-231 cells between ARID1A^NC^ and ARID1A^KO^ groups. (**B**) Western blot assay of cytoplasmic and nuclear YAP and p-YAP expression in MDA-MB-231 cells between ARID1A^NC^ and ARID1A^OE^ groups. (**C**) The subcellular localisation of YAP in MDA-MB-231 cells between ARID1A^NC^ and ARID1A^KO^ groups. (**D**) Western blot and Co-IP assays were performed to assess the interaction between ARID1A and YAP. (**E**,**F**) ARID1A, YAP, BRG1, and TEAD interaction were detected in MDA-MB-231 cells using anti-ARID1A, anti-YAP, anti-BRG1, and anti-TEAD antibodies for Co-IP and Western blot assays. Error bars represent the standard deviation among replicates. Data are presented as the mean ± SEM (n = 3). The uncropped blots are shown in Appendix A.

**Figure 7 cancers-15-02447-f007:**
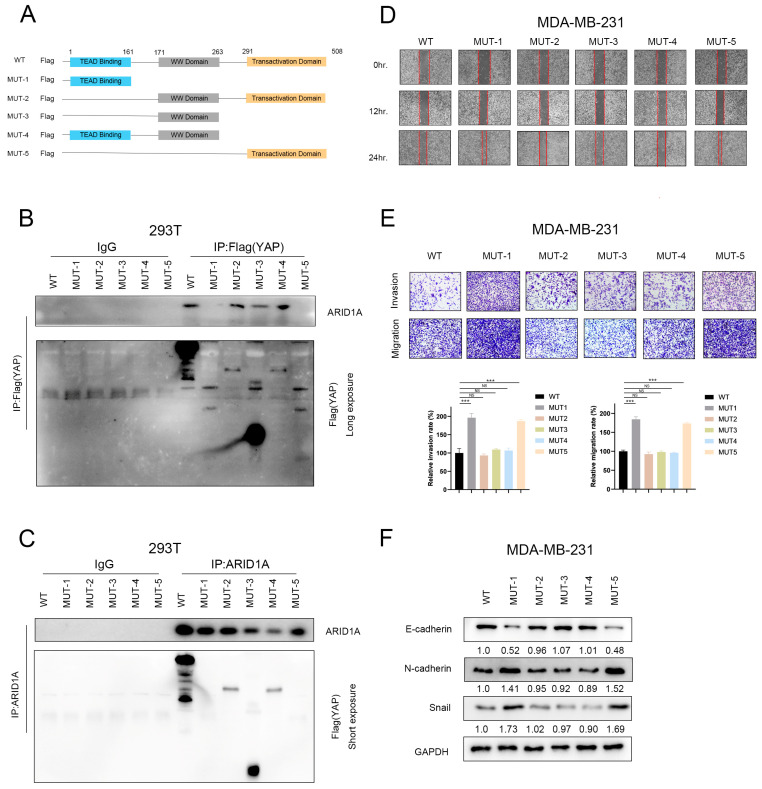
ARID1A combined with YAP via the WW domain. (**A**) Schematic diagram of the truncated construct of YAP. (**B**,**C**) HEK293T cells were transfected with the indicated plasmids. Western blot and Co-IP assays were performed to assess the interaction between ARID1A and YAP-truncated mutations. (**D**,**E**) Migration and invasion of MDA-MB-231 cells of each group detected by wound healing assay (**D**), transwell migration assay and transwell invasion assay (**E**). (**F**) Western blot analysis of EMT markers expression in each group of MDA-MB-231 cells. Error bars represent the standard deviation among replicates. Data are presented as the mean ± SEM (n = 3), NS indicates not significant, *** *p* < 0.05. The uncropped blots are shown in Appendix A.

**Figure 8 cancers-15-02447-f008:**
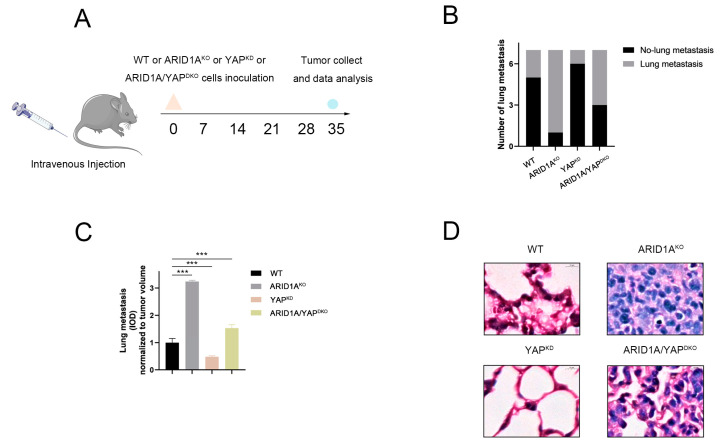
ARID1A and YAP co-regulate tumour metastasis in vivo. (**A**) Schematic diagram of a xenograft tumour model. (**B**) Incidence of lung metastasis in ARID1A^NC^, ARID1A^KO^, YAP^KD^, and ARID1A/YAP^DKO^ cells. (**C**) Quantitation of tumour foci (IOD of fluorescent area) in the lung of mice injected ARID1A^NC^, ARID1A^KO^, YAP^KD^, and ARID1A/YAP^DKO^ cells; (**D**) Hematoxylin and eosin (H&E) staining analysis of tissue sections from recipient mice at 5 weeks after transplantation. Scale bar, 10 μm. Error bars represent the standard deviation among replicates. Data are presented as the mean ± SEM (n = 5), *** *p* < 0.05.

**Figure 9 cancers-15-02447-f009:**
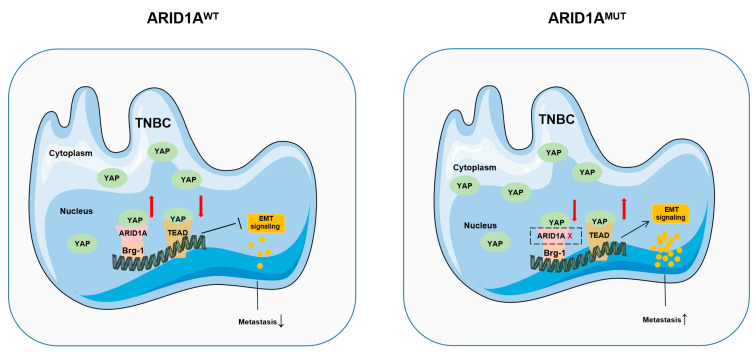
ARID1A regulates tumour metastasis by competitively binding YAP to form an ARID1A/YAP complex. Under normal conditions, ARID1A promotes the entry of YAP into the nucleus to form the ARID1A/YAP complex and inhibits the formation of the YAP/TEAD complex, thereby inhibiting the expression of EMT and affecting tumour metastasis. In contrast, deletion of ARID1A increases the formation of the YAP/TEAD complex and promotes the expression of EMT, thereby increasing tumour metastasis. ↑ means promote or increase, ↓ means inhibit or decrease, and × means the absence of ARID1A.

## Data Availability

RNA-seq is currently being uploaded in the Gene Expression Omnibus (GEO) repository.

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
