# Peer review of "Chromatin Remodelling Molecule ARID1A Determines Metastatic Heterogeneity in Triple-Negative Breast Cancer by Competitively Binding to YAP"

_cancers, 2023, doi:10.3390/cancers15092447_

Round 1

Reviewer 1 Report

Wang et al. focused on the role of the gene ARID1A in triple-negative breast cancer. The results indicate that ARID1A could play an important role in cell migration and invasion via the Hippo/YAP signaling axis.

In general, I think the authors did quite some work, and the results are convincing and interesting. The analysis pipeline that the authors used is sophisticated and impeccable. Although some details need further explanation, I have some questions related to the methods and results. Major revisions are needed. Here are some questions and comments that the authors should think about as they make changes to their work.

  1. The authors' research revealed a substantial difference in EMT between YAP-KO and YAP-NC cell lines. YAP was shown to colocalize differently in the cytoplasm and nucleus in the ARID1A-NC and ARID1A-KD cell lines. HeLa cells exhibit heterogeneous cell growth, according to Tao Li et al (PMC8288455). They noticed that a cell with a high density of YAP lost nuclear localization displayed a faster rate of proliferation, which displayed the same trend as the ARID1A-KD cells in this study. And in vivo, ARID1A-KD cells showed increased tumorigenicity. These cells' rates of cell division, however, were not assessed. As a higher growth rate might produce more obvious clones, the variable tumorigenicity of these cells in the mouse model might be subjected to a distinct cell proliferation rate. In the amended text, the authors should either discuss or add relevant results.
  2. It is incorrect to use mice tumorigenicity studies to highlight variations in tumor metastatic capacity. The author injected the cells into the mice through the tail vein, as described in the Materials and Methods section, and then counted the tumor clones to show the variation in the ability of tumor cells to metastasize. The amount of clonal clusters is unrelated to metastasis and can only be used to compare the tumorigenicity of various cell types. Mice's blood-injected tumor cells resemble circulating tumor cells more closely. The authors' tumor cells were injected into mice, not naturally released into the circulatory system by the tumor, despite the fact that circulating tumor cells and tumor migration are strongly associated. As a result, the results drawn from this collection of tests contain mistakes. Tumor metastasis is not correlated with the frequency of clonal clusters.

Additionally, there are a few minor issues:

  1. It is difficult to determine the precise date of the selection of stably transfected cells in lines 94–95. Also, a week seems insufficient to choose the stably transfected cells.
  2. The relevance of morphological differences was only weakly supported by Figure 2B.
  3. Seven mice seem to be insufficient in mouse trials. 
  4. Figure 3E must contain the images corresponding to the ARID1A-NC tumor locations.
  5. Given that migration and invasion should have decreased due to the YAP-KD, the labels in the second column of Figure 5F may be incorrect. Nevertheless, YAP-KD cells' increased invasion and migration were visible in Figure 5F's second column.
  6. There was no mention of the HEK293T cell resource.
  7. Languages ought to be enhanced. There were errors, such as 1X105 in line 174 and H2O2 in line 184.

Reviewer 2 Report

The authors in this paper described how the ARID1A/YAP interaction determines metastatic heterogeneity in triple negative breast cancer (TNBC). In the introduction they highlighted the importance of searching and individualising biomarkers as target for the treatment of metastatic TNBC since the targets used up to know have limited efficacy. They introduced and described the role of ARID1a (supported by enough literature) in different type of cancer with a lack in the metastatic TNBC justifying in this way the main aim of their project. Based on their pervious evidence that suggests ARID1a plays a role in the invasive metastatic TNBC in this paper they develop experiments to prove it, individualising as crucial the interaction with YAP/Hippo pathway. The experiments done lead them to the conclusion that ARID1a binds YAP inhibiting the production of the YAP/TEAD complex preventing their transcriptional activating and consequently tumour proliferation and metastasis. The rational in support of their idea is well described but they could have add some more references (Chao Tang, et al. , Front.Oncol., 04 Oct 2021; Anwesha Dey et al., Cells 2022, Jul; 11) to highlight how even though the interaction of ARID1A with YAP, that prevents its binding with TEAD, has been described, their effect in metastatic TNBC has not been described yet as well as the role of ARID1a in this tumour type.

All the experiments done, the material and methods and the results need to be better explain and clarify. My suggestions are the following:

The third paragraph of Materials and Methods (Lentiviral sRNA production) needs to be clarify, it’s not clear where and when the CRISP/Cas9 method and when the shRNA have been used. They are two different techniques; the result of the CRISP/Cas9 in a knockout, the result of shRNA is knockdown and also in the figures they always referred to sh-ARID1A or ARID1A KO with no enough explanation. Which plasmid has been used for the overexpression of ARID1A, both for the human and murine cell lines?

In the fourth paragraph (Cellular fractionation, WB, Co-IP) is not explained how  the nuclear and cytoplasmic protein extraction has been done, nothing is mentioned in the material and methods even though they claim the different level of YAP  in the nuclear vs cytoplasm is important evidence here.

In the Immunofluorescent staining paragraph, is the antibody for YAP the same mentioned for the WB? Add the cat number.

In the Immunohistochemical analysis paragraph how many samples (patients) have been used? Where this samples arrived from? Hospital/Trial?

In the results the Fig 1 is fine as well as the corresponding paragraph.

Results 3.2 needs clarification, since the CRISP/Cas9 technique has been used I think it’s better to use ARID1A KO as nomenclature/legend across all the experiments done and in the figures (shARID1A to me means silencing with siRNA). Speaking about silencing, inhibition and knockout for me it’s confusing since they are not synonymous. Is not clear if a silencing or a knockout has been performed and in which experiment. Better explain this please. Justify the choice of the MDA-MB-231 cell liens. In the figure 2A what the sh-NC (is just the Cas9 as control or the cell line) is? The same for OE-NC, is the empty plasmid control or the MDA-MB-231 cell line? In my opinion all the controls need to be showed in a WB, MDA-MB-231(wildtype)/MDA-MB-231 Casp9 or empty plasmid/ then the MDA-MB-321 KO or OE.  Mainly because the level of ARID1A in the sh-NC in Fig 2A looks higher than the one if fig 1f, it’s probably related to the different WB run at different time and separately but that’s why I think is fundamental to have all the controls in the same gel and eventually show them in the Supplementary. Please provide this western blot. In figure 2B-2C-2D NC has been used to address the wild type, that let me assume that in fig 2A shNC and OC-NC are the control for the transfection? Explain this please, and better clarify the reason behind the use of sh-ARID1A or/and ARID1A KO since in the material and method and in the results it’s not clear. Again, why have been used sh-ARID1A-1/sh-ARID1A-2 and ARID1AKO? If there are differences and different methods have been used in different experiments is not clear.

In the result 3.3 the in vivo experiment is not clear, in the material and methods the murine cell line 4T1 has been described as the one used for the experiment but in the result paragraph the authors mention the human cell line MDA-MB-231, explain this please. Also in this case a better clarification of the ARID1A KO and ARID1A NC.  Also because in the material and methods two gene sequences are described for the mouse experiment, which one has been used? I assume that the choice of the lung as organ to analyse, it is due to the fact is the second organ, after the bone, breast cancer metastasizes.  

Results 3.4 just include the number of tumour samples analysed.

In the result 3.5 YAP has been silenced or knockout? How?

In the result 3.6 the housekeeping controls have been swapped in Fig 6A-B and S2B-C, since the Histone H3 is nuclear and GAPDH cytoplasmic. The figure 6D and S2D, is not clear if the Co-IP has been done on nuclear extract (vague explanation in the results)or total extract, to me this Co-IP shows that ARID1A and YAP co-precipitate, so interact;, since they immunoprecipitated ARID1A and did a WB for YAP is obviously that they observed low YAP in the shARID1A and more YAP in the ARID OE, simply because they precipitate more (in the OE) and less (in the sh) ARID1A. The sentence “Moreover, YAPS forms a compound with BRG1” needs a reference.  The knockout (or silencing) of BRG1 has never been mentioned before, neither in the material and method, just appear in figure 6E. Justify this, why did you perform it and how? In this way the figure 6E is not clear.

The results 3.7 needs to be better explain the experiment performed is not clear, why in this case a different cell line has been used? The Flag (Yap) mentioned in the Co-IP (Fig7B) is part of the plasmin use in Fig 7A? How do you have a Flag-YAP in this cell line?

Results 3.8. Which cell line has been use in this experiment?

It’s difficult to review the results the authors obtained since most of their experiments have not been explained properly, is not clear what they do and a bit confusing. They put a lot of effort in this project and a lot of experiments have been done to proof they hypothesis but a few controls are missing and they need to better justify the reason behind heir experiments (material and methods and rationale).

Round 2

Reviewer 1 Report

The authors responded to all of my concerns carefully. I have no further questions.

However, the non-standard writing of professional vocabulary still exists. For example, H2O2 in line 184. Languages ought to be enhanced. Besides, the reselution of all figures in the revised manuscript was too low to identify valid information.

Since the impact of cell proliferation on tumorigenicity experiments has been considered, the discussion section should also cite relevant literature, such as PMC8288455, etc., to discuss whether ARID1A plays a role in the correlation between YAP nuclear cytoplasmic co-localization differences and cell phenotype.

Reviewer 2 Report

Dear authors, thank you for all you corrections and explanations, everithing is clear and very well explained.

Author Response

Dear Reviewer,

We sincerely thank the reviewer for your valuable feedback that we have used to improve the quality of our manuscript entitled ‘Chromatin remodeling molecule ARID1A determines metastatic heterogeneity in triple-negative breast cancer by competitively binding to YAP’ (cancers-2278291).

We would like to thank the referee again for taking the time to review our manuscript.